# Acute effect of transcutaneous auricular vagus nerve stimulation on cardiac vagal activity in men living with HIV: A proof-of-concept clinical trial

Jason Azevedo de Medeiros[1]*, Uirassu Borges[2], Phelipe Wilde[1],
Rafaela Catherine da Silva Cunha de Medeiros[3], Júlio César Medeiros Alves[1],
Amon Gonçalves de Melo Neto[1], Jason R. Jaggers[4], Daniel Gomes da Silva Machado[1],
Ronaldo Vagner Thomatieli dos Santos[5], Paulo Moreira Silva Dantas[1]

**1** Center for Health Sciences, Federal University of Rio Grande do Norte, Natal, Rio Grande do Norte,
Brazil, **2** Institute of Psychology, German Sport University Cologne, Cologne, Germany, **3** Department
of Physical Education, State University of Rio Grande do Norte, Mossoró, Rio Grande do Norte, Brazil,
**4** Department of Health and Sport Sciences, University of Louisville, Louisville, Kentucky, United States
of America, **5** Department of Biomedical Sciences, Federal University of São Paulo, Santos, São Paulo,
Brazil

\* jason.medeiros1@hotmail.com

pone.0326793

TÜRKIYE

**Peer Review History:** PLOS recognizes the
benefits of transparency in the peer review
process; therefore, we enable the publication
of all of the content of peer review and
author responses alongside final, published
articles. The editorial history of this article is
available here: https://doi.org/10.1371/journal.
pone.0326793

## Abstract

This proof-of-concept study evaluated the acute effects of transcutaneous auricular
vagus nerve stimulation (taVNS) on cardiac vagal activity in people living with HIV.
Twenty-one men living with HIV on antiretroviral therapy participated in a single-blind,
crossover clinical trial. Participants underwent two counterbalanced stimulation
conditions (taVNS and sham) with a 48-hour washout period. Cardiac vagal activity
was assessed using vagally-mediated heart rate variability (vmHRV) indices, includ-
ing the root mean square of successive differences (rMSSD) and the percentage of
differences between adjacent normal intervals greater than 50 ms (pNN50), recorded
before, during, and after stimulation. No significant changes in vmHRV parameters
were observed over time or between conditions. These findings suggest that an acute
taVNS session does not modulate cardiac vagal activity in people living with HIV. We
discuss potential explanations for these results and highlight considerations for future
research on taVNS as a non-pharmacological approach to autonomic modulation.

## Brazilian registry of clinical trials

RBR-8k54cz.

## Introduction

People living with human immunodeficiency virus (PLHIV) may present compromised
cardiac vagal activity due to the virus's infection in different body systems, including

**Data availability statement:** The data from this study can be accessed through the following link: https://doi.org/10.7910/DVN/DM4G9A.

**Funding:** This study was partially funded by the Coordination for the Improvement of Higher Education Personnel (CAPES) - Brazil (Funding Code 001), which awarded doctoral scholarships to JAM and PW. https://www.gov.br/capes/pt-br PMSD also received support from the National Council for Scientific and Technological Development (CNPq) through a government grant (409050/2021-0). https://www.gov.br/cnpq/pt-br. The funders had no role in study design, data collection and analysis, decision to publish, or preparation of the manuscript.

**Competing interests:** The authors have declared that no competing interests exist.

the central nervous system, autonomic nervous system, and cardiovascular system [1]. The brain-heart axis can experience communication issues due to metabolic dysfunction induced by HIV attacking specific brain regions responsible for cardiovascular adjustments, such as heart rate and blood pressure control [1]. The primary method of controlling the virus is through continuous antiretroviral therapy, which enables better control of viral load and immune system function, reducing mortality rates and improving patient's life expectancy and quality of life [2–5]. However, since it is a long-term treatment, antiretroviral therapy can lead to adverse effects, including the development of insulin resistance, dyslipidemia, and lipodystrophy [6]. These effects may exacerbate the impairments caused by HIV's direct impact on the central nervous, autonomic, and cardiovascular systems, thereby increasing the risk of developing cardiovascular disease [1,6–9]. Transcutaneous Auricular Vagus Nerve Stimulation (taVNS) is a non-pharmacological adjunct therapy with the potential to help mitigate some of the long-term adverse effects of antiretroviral therapy, particularly regarding cardiac autonomic function [10]. The taVNS is a non-invasive technique that uses electrical stimuli to modulate vagus nerve activity, thereby influencing brain activity through vagal afferent pathways [11]. This technology has been widely used in research and therapy due to its safety and lack of significant side effects [12]. However, there are currently no studies that address the use of taVNS directly related to HIV. This study addressed this issue with the aim of investigating the influence of taVNS on cardiac vagal activity in PLHIV. This proof-of-concept study seeks to provide preliminary insights on the feasibility and potential acute effects of taVNS on cardiac vagal activity in PLHIV, setting the foundation for larger-scale studies on non-pharmacological interventions for autonomic regulation in this population.

Essentially, the invasion of HIV into the central nervous system can be explained by some hypotheses: the "Trojan horse theory," in which the virus uses monocytes or CD4 + T lymphocytes as a means of transport [13]; crossing of the blood-brain barrier through the endothelial cells [14], or by cell transfer facilitated by the gp120 protein [15]. Additionally, HIV can affect the central nervous system through retrograde axonal transport, particularly via the gp120 protein [16]. This mechanism is associated with neurodegeneration and neuronal apoptosis, contributing to the neurological complications observed in patients with HIV [17]. In the brain, HIV can cause metabolic disturbances affecting the hypothalamic-pituitary-adrenal axis and the autonomic nervous system, leading to abnormal levels of adrenaline and poor regulation of the cardiovascular system, including changes in heart rate variability (HRV) [18]. In accordance, lower resting HRV indices in people with HIV has been associated with lower cardiac vagal activity compared to people without the HIV [19,20]. In this context, the chances of developing cardiovascular diseases due to HIV become higher. Thus, non-pharmacological therapies focused on autonomic adjustments could potentially be beneficial in reducing cardiovascular risks, as is the case with taVNS.

The mechanism of action of taVNS on cardiac vagal activity can be explained by the presence of vagal nerve endings in the human auricle, especially in the regions of the cymba conchae and tragus. When stimulated, these endings send signals to the nucleus tractus solitarius, which projects to different areas of the brain, such as

the prefrontal cortex [21]. According to the neurovisceral integration model, the prefrontal cortex regulates cardiac function through cardiac vagal activity [22]. Therefore, the application of taVNS in people with HIV seems promising because it targets a mechanism of action in the brain areas infected by the virus. Studies demonstrate beneficial effects of taVNS in healthy populations [23,24] as well as in groups with clinical conditions such as depression, epilepsy, obesity, and cardiovascular diseases (heart failure, hypertension, and atrial fibrillation) [23,25–31]. However, despite major impairment of cardiovascular activity caused by HIV [32], there is a lack of knowledge regarding the effect of taVNS on any clinical outcome in PLHIV.

Understanding how taVNS can modulate cardiac autonomic function allows for a potential application of taVNS in reducing cardiovascular risks in people living with HIV, who often exhibit diminished cardiac vagal activity as a consequence of HIV infection and antiretroviral therapy. These autonomic dysfunctions are associated with an increased cardiovascular risk, impacting on the quality of life and longevity of this population. Given that taVNS may positively influence vagal tone, this study aimed to investigate the acute effect of taVNS on cardiac vagal activity in men living with HIV (MLHIV). Therefore, this exploratory investigation assesses whether taVNS can induce an acute vagal response, suggesting its potential as a non-pharmacological intervention to improve autonomic function and reduce cardiovascular risks in PLHIV.

## Materials and methods

The study is characterized as a randomized single-blind clinical trial, designed as a proof-of-concept investigation. The study was restricted to male participants for three reasons: First, sex differences in heart rate variability are evident and this could make the results confusing [33]. Second, there is a difference in the autonomic response between the sexes during electrical stimulation of the vagus nerve [34]. Third, women can show fluctuations in HRV, with notable changes between the follicular and luteal phases, mainly due to variations in progesterone levels [35,36]. Thus, the sample consisted of 21 men living with HIV [age: 39.5 ± 11.68; Body Mass Index (BMI): 23.5 ± 4.29 kg/m$^2$] who had been on antiretroviral therapy for a minimum of 6 months. A priori sample size calculation was conducted using effect size f = .24 for repeated measures analysis of variance with between factors based on previous study [37]. The following input parameters were adopted: α error probability of.05, power of.80, two number of groups, six number of measurements, and correlation among repeated measures of 0.5. The total sample size calculated was defined as 20 participants. The sample size calculation was performed using the software G*Power [38].

Participants were recruited from a referral hospital for infectious diseases and the Specialized HIV Care Service (SAE-HIV) through announcements in healthcare settings and via telephone contact. Participants who had been part of the "Viver Mais" project (an extension project at UFRN that offers physical exercise and nutritional guidance for people living with HIV) were also contacted. Recruitment began on July 1, 2022, and ended on August 30, 2022. All participants were informed about the risks and benefits of participating in the study by reading the informed consent form. After agreeing to participate, participants were required to sign the form in writing. The study was approved by the ethics committee of the Onofre Lopes University Hospital of the Federal University of Rio Grande do Norte – HUOL/UFRN (approval number: 3.360.663), and pre-registered in the Brazilian Registry of Clinical Trials (Registration Number: RBR-8k54cz). Additionally, this study followed CONSORT recommendations [39].

This investigation initially aimed to examine the chronic effects of taVNS combined with physical exercise on various parameters, including metabolic, cardiovascular, and cognitive measures. However, the COVID-19 pandemic and subsequent laboratory closures led to unforeseen circumstances and certain technical limitations, prompting us to shift our focus to explore the acute effects on cardiovascular parameters, specifically on cardiac vagal activity. Additionally, despite extensive efforts to recruit participants in hospitals and collaborate with infectious disease specialists, we encountered significant challenges in achieving a larger sample size. A total of 35 individuals were assessed for eligibility; however, 14 did not meet the inclusion criteria, resulting in a final sample of 21 participants. While we acknowledge that a larger sample

size could provide more robust estimates, we believe that the data obtained contribute to a better understanding of the potential acute effects of transcutaneous auricular vagus nerve stimulation (taVNS) in PLHIV, serving as a foundation for future investigations.

Inclusion criteria required subjects to be diagnosed with HIV/AIDS [40], asymptomatic, and free of opportunistic infections. They needed to have been on ART for at least 6 months, be between 18 and 50 years old, and not present cardiopathies. Exclusion criteria included the absence of facial or ear pain, recent auditory trauma, metallic implants, including pacemakers, personal or family history of seizures, mood or cardiovascular disorders, alcohol dependence, recent use of illicit drugs, smoking, or use of any pharmacological agents known to increase the risk of seizures (Fig 1) [11].

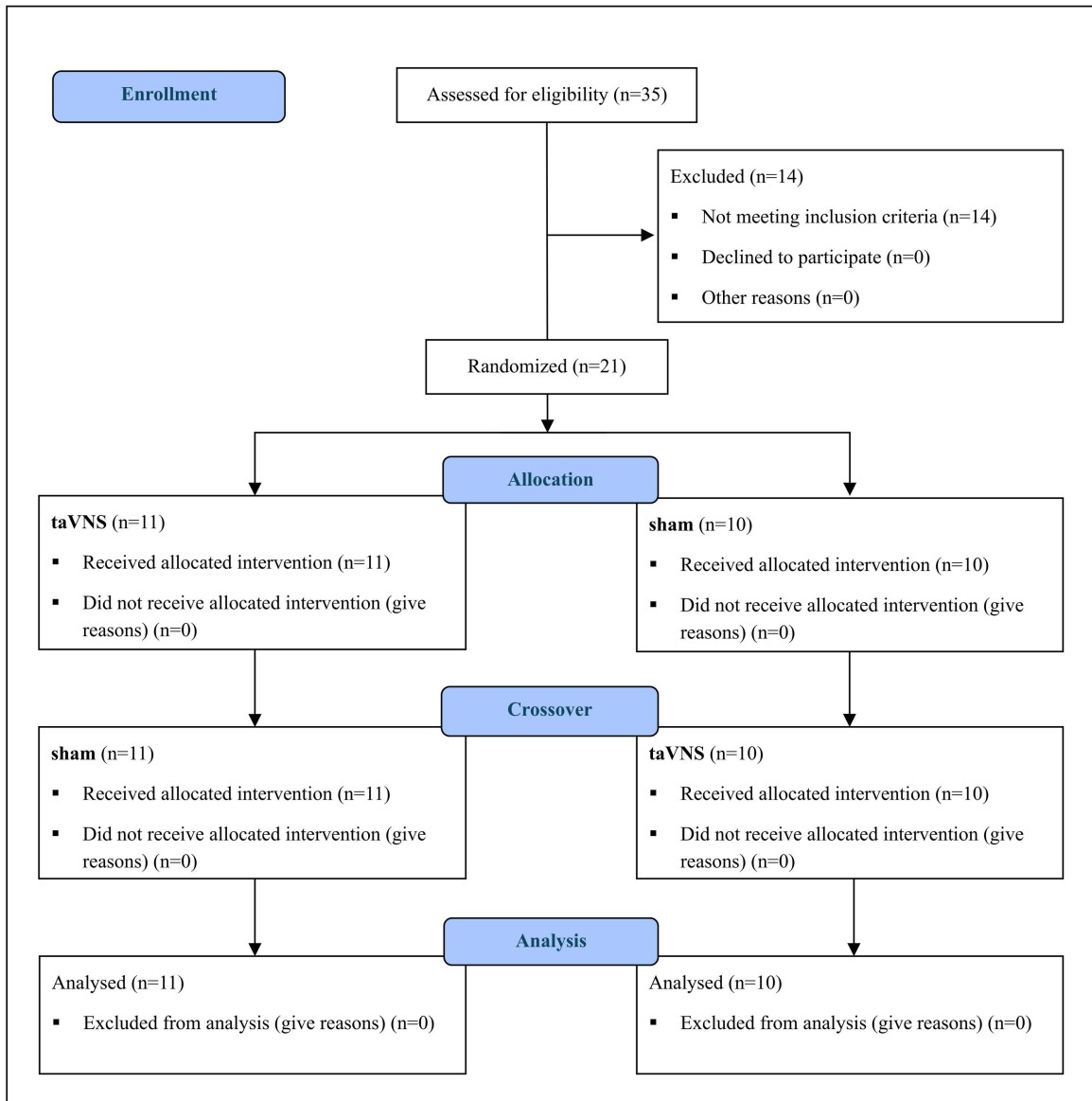

**Fig 1. Study flowchart.**

## Transcutaneous auricular vagus nerve stimulation

For taVNS, a transcutaneous electrical nerve stimulation (TENS) device (Neurodyn Portable – 2 Channels – TENS) was utilized. Tin electrodes were attached to a 3D-printed headphone-like attachment with adjustable settings for a better fit in the cymba concha of the left ear (Fig 2) [21]. The choice of the left ear for auricular stimulation of the vagus nerve was supported by anatomical and functional evidence indicating a greater density of innervation, significant neuroprotective effects and greater consistency in the therapeutic response [21,41,42]. In addition, for greater comparability we followed the consensus document by Farmer et al. [11] which highlights the predominance of stimulation protocols in the left ear.

To remove oil and reduce skin surface resistance, a compress with 70% isopropyl alcohol was applied to the auricular area designed for stimulation. A thin layer of conductive gel was then evenly spread on the electrode surface before securing it to the participant's cymba conchae. The technique, targeting, and considerations for laboratory administration of taVNS and sham followed recommended guidelines [43]. Stimulation was applied in the morning, with the participant resting in a seated position for 30 minutes. Continuous stimulation has standardized with a pulse width of 500 µs and a frequency of 10 Hz. The stimulation intensity was set at 200% of the individual perceptual threshold.

To determine the perceptual threshold, a trained researcher attached the electrodes to the participant's left ear and then gradually adjusted the intensity of the stimulator, starting at 0 mA. The intensity was increased or decreased until the participant reported a slight sensation of vibration. The perceptual threshold was confirmed after the participant gave four consecutive positive responses to the same intensity. This procedure was standardized and applied in both conditions, taVNS and sham. After identifying the perceptual threshold, the stimulator was turned off to calculate the percentage of intensity to be applied during the session. At this point, the participant was informed that the stimulation session would last 30 minutes. In the taVNS condition, the stimulator was turned on, and an intensity corresponding to 200% of the perceptual threshold was applied. In the sham condition, the electrodes remained in place, but the stimulator was turned off. The choice of the sham protocol is in line with previous studies that used the same ear region as the active condition to position the electrodes while the electrostimulator remained switched off [42,44,45]. There are protocols that use the ear

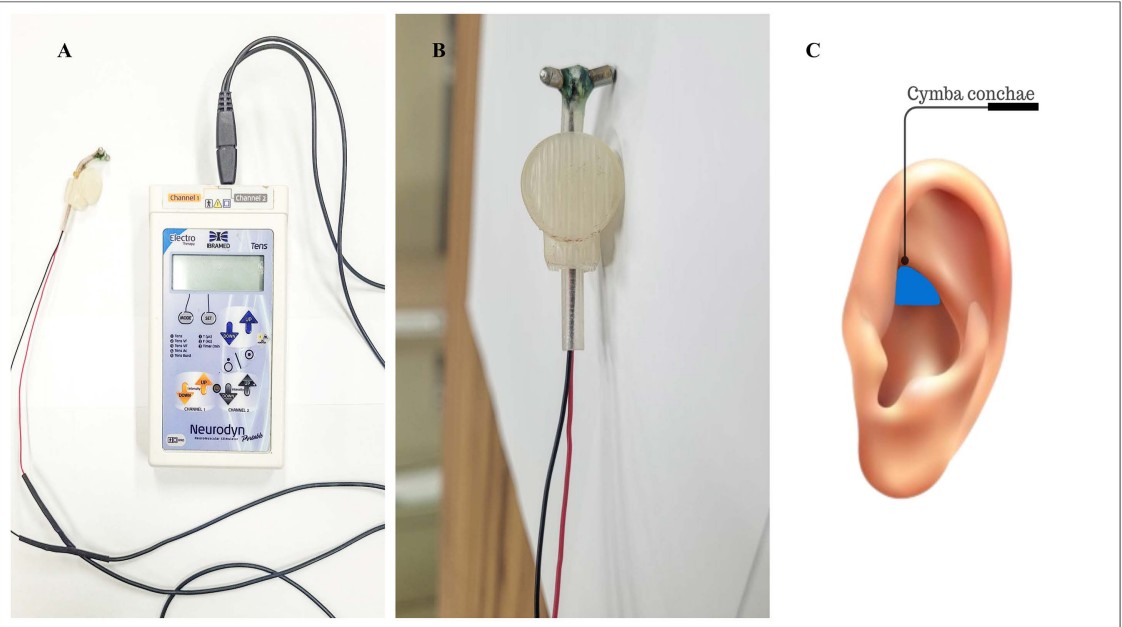

**Fig 2. Stimulation device (A), modified electrode (B), and region on the auricle where the electrodes were positioned for stimulation (C).**

lobule as an active sham [11]. However, it is possible that stimulation in this region may not be entirely specific and could influence multiple neural pathways, leading to activation patterns similar to taVNS stimulation [21].

In both conditions (taVNS and sham), the researcher remained beside the stimulator throughout the session to intervene in case any participant reported discomfort. During the taVNS condition, intensity adjustments were made for four participants who reported discomfort: one participant had the intensity reduced to the perceptual threshold, while the other three had the intensity adjusted to levels still above the perceptual threshold (118%, 125%, and 162%, respectively). In the sham condition, although the stimulator was turned off, two participants requested a reduction in intensity because they believed they were feeling stimulation.

To minimize potential bias in blinding, all participants were informed in both conditions that the stimulator would remain on during the session, even if they expressed doubts about perceiving the stimulus.

## Cardiac vagal activity and cardiovascular parameters

Cardiac Vagal Activity was assessed through HRV. Participants were advised not to consume energy drinks, alcohol, or coffee, avoid intense physical exercise, plan for a good night's sleep, and abstain from eating in the two hours before the protocol. All participants confirmed compliance with these criteria before the experimental assessments.

Upon arrival at the laboratory, participants were equipped with the Polar H10 heart rate transmitter (Polar Electro Oy, Kempele, Finland), known for its high-quality RR interval signal [46] and strong agreement with ECG records for HRV analysis [47].

Then, participants were seated on a comfortable couch with feet on the ground, knees flexed, and without crossing their legs for a 10-minute period to stabilize heart rate. They were instructed to refrain from talking, sleeping, or making sudden movements during the evaluation. Following this, the heart rate monitor was connected to the Elite HRV smartphone app. RR intervals were recorded for 10 minutes during the rest period. The recording was then saved, and the device was temporarily stopped to identify each participant's stimulation perception threshold. Subsequently, with either taVNS or sham stimulation, RR intervals were recorded continuously. The recording lasted for 90 minutes: 30 minutes of stimulation and an additional 60 minutes of recovery.

The RR interval sequence was saved as text files, anonymized by another researcher who modified file names, and later imported into the Kubios HRV Premium software version 3.5.1 (Biosignal Analysis and Medical Imaging Group, Department of Physics, University of Kuopio, Kuopio, Finland) for analysis [48]. The Kubios software was configured to apply low-level artifact correction (0.35 seconds). The algorithm uses a threshold-based detection method, comparing each RR interval to the local average interval calculated through median filtering of the RR interval time series. This approach ensures that the local average is not influenced by outlier RR intervals. RR intervals that deviate from the local average beyond the specified threshold are identified as artifacts and marked for correction by the software [49]. In the present study, the number of corrected heartbeats was less than 2%, as recommended by the Kubios guide, which advises that the percentage of corrected beats should be < 5% to avoid significant distortion (suppressed variability) in the analysis results [49]. This process minimizes errors caused by ectopic beats or signal noise, ensuring a cleaner and more accurate dataset for HRV analysis. This approach aligns with the best practices for artifact correction in HRV studies [50,51].

A total of six 10-minute segments were extracted from the timeline for analysis: Moment 1 (baseline), Moments 2 and 3 during stimulation (0–10 min and 20–30 min), and Moments 4, 5, and 6 during recovery (30–40 min, 50–60 min, and 80–90 min). Within each 10-minute window corresponding to these moments, HRV was analyzed in the first 5 minutes across both conditions and for all participants to ensure better data comparability.

The chosen HRV indices were the root mean square of successive RR interval differences (rMSSD in ms) and the percentage of differences between adjacent normal intervals (pNN50 in %) as they better reflect cardiac vagal activity [50,51].

Systolic and diastolic blood pressures (SBP and DBP) were measured after 10 minutes of rest at baseline using an Ambulatory Blood Pressure Monitor CONTEC® ABPM50 (Contec Medical Systems – Hebei, China).

**Procedure.** The participants attended three laboratory visits, each separated by 48 hours. During the initial visit, they participated in a structured anamnesis interview, which included questions about age, current antiretroviral therapy, diagnosis time, and the submission of a physician-authorized clinical report containing data from the most recent CD4 and viral load tests. Subsequently, body composition was assessed. Weight and height measurements were obtained using a digital scale with an integrated stadiometer (Welmy®), calibrated in accordance with the manufacturer's guidelines. Measurement protocols adhered to the standards established by standards for anthropometry assessment [52]. Body fat percentage and fat-free mass were assessed using dual-energy X-ray absorptiometry (DEXA) [53] with a GE Lunar Prodigy Advance 2015 device (GE Medical Systems, Madison, WI), and analyzed via Encore Version 15 software. Based on trunk and lower limb fat percentage values, participants were classified regarding the presence or absence of lipodystrophy using cutoff points proposed by Mialich et al. [54]. Randomization was performed by a blinded researcher using randomizer.org, where subjects were divided into two conditions (taVNS and sham) in a counterbalanced manner. Subjects were crossed over after the first stimulation, with a minimum washout of 2 days between sessions.

On the second and third visits, participants were subjected to either the experimental or sham protocol. Once seated on a comfortable couch, they were presented with an 11-point visual analog scale and instructed to rate their pain perception. They were then asked to choose a video to watch for the duration of the experimental session. The videos exceeded the session length (>120 min) and featured point-of-view (POV) recordings of walks through various countries, showcasing well-known landmarks. The videos contained no audio or content that might evoke emotions such as anger, sadness, or euphoria. Following the selection of the video, RR interval recording commenced for a 10-minute baseline period. The recording was paused to determine the perception threshold. Subsequently, RR intervals were continuously recorded throughout the entire stimulation period (30 min) and recovery period (60 min). At the end of the stimulation period, the earbud-shaped electrode was removed from the participant's ear, and the investigator presented the visual analog scale to the participant. The timeline of the stimulation session can be better visualized (Fig. 3).

### Evaluation of safety and tolerability

Throughout the stimulation, regardless of the condition, participants were monitored for potential adverse events, including extreme reductions in heart rate to less than 35 beats per minute, respiratory difficulty, cutaneous discomfort, irritation, headache, facial pain, and dizziness. No adverse events were reported. A visual analog pain scale with a numerical rating

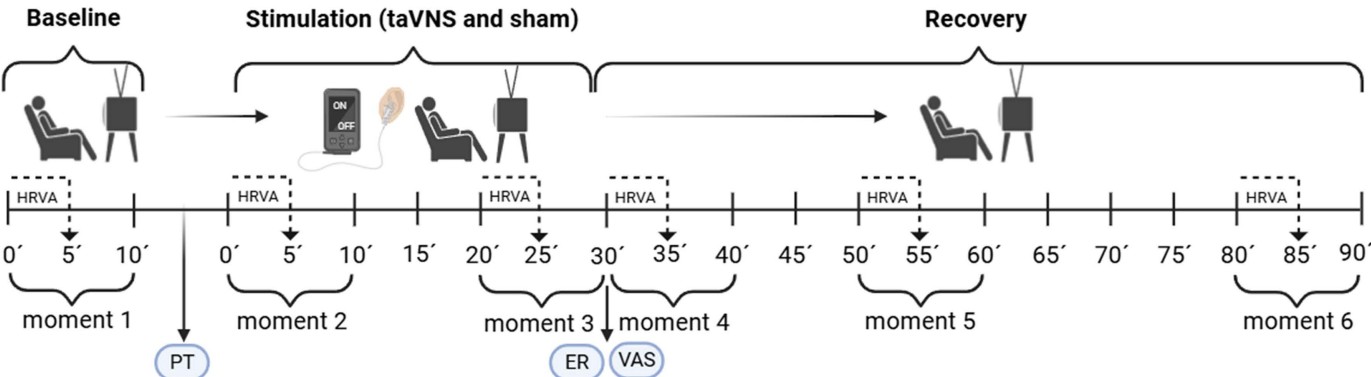

**Fig 3. Timeline of recording and analysis of RR intervals.** HRVA, heart rate variability analysis; PT, perception threshold; ER, earphone removal; VAS, visual analog scale.

scale ranging from 0 to 10 was presented to each participant after both stimulation conditions. "0" was used as the lowest score for no perceived sensation, and "10" was adopted as unbearable pain.

## Statistical analysis

For assumption testing, Q-Q plots of residuals were generated and inspected, and the Shapiro-Wilk normality test was conducted. A t-test was performed to compare baseline variables (SBP and DBP) between the taVNS and sham conditions, with effect sizes calculated using Cohen's d. A Generalized Estimating Equation (GEE) model for fixed effects (condition, time, condition and time interaction) with a gamma distribution and an identity-link function was selected for the analysis of rMSSD and pNN50, as these variables exhibited a gamma distribution with right-skewness [55,56]. The unstructured covariance matrix was adopted for analysis [57]. It was defined based on the lowest Akaike Information Criterion (AIC) compared to Gaussian distribution. Additionally, residuals were examined to ensure the adequacy of the model assumptions. Effect sizes (Cohen's d) were calculated for both within- and between-condition comparisons. For between-condition comparisons, Cohen's d for groups of equal size was used. For within-condition comparisons across time points, Cohen's d for repeated measures was used and reported as "d repeated measures, pooled" [58]. All analyses were performed using Jamovi (Version 2.4), with a significance level set at p < .05 [59–61].

## Results

The characterization of participants, including age, body composition, and clinical parameters associated with the diagnosis and treatment of HIV, is presented in Table 1.

Comparisons of autonomic and cardiovascular parameters and response variables to stimulation were made at baseline between conditions. No differences were found for perception threshold, indices HRV, systolic blood pressure (SBP),

**Table 1. Characterization and clinical parameters of the participants.**

| | N | Mean (SD) | 95% Confidence interval | |
| --- | --- | --- | --- | --- |
| | | | Lower limit | Upper limit |
| Age (years) | 21 | 40.1 (11.5) | 36.4 | 43.7 |
| Height (cm) | 21 | 168.8 (7.0) | 165.6 | 172.0 |
| Body weight (Kg) | 21 | 66.4 (12.6) | 60.7 | 72.1 |
| BMI (kg/m²) | 21 | 23.5 (4.3) | 21.5 | 25.4 |
| Fat (%) | 21 | 27.4 (6.7) | 24.4 | 30.5 |
| FFA (kg) | 21 | 48.3 (7.0) | 45.1 | 51.5 |
| Diagnosis time (years) | 21 | 12.3 (7.4) | 9.59 | 14.9 |
| Time to cART (years) | 21 | 10.7 (6.8) | 8.28 | 13.2 |
| Viral Load | 21 | Undetectable | ------- | ------- |
| **cART and Lipodystrophy** | | (%) | | |
| II+NRTI | 21 | 11 (52.38%) | | |
| NRTI+NNRTI | 21 | 7 (33.33%) | | |
| PI+NRTI | 21 | 1 (4.76%) | | |
| IP+NNRTI | 21 | 1 (4.76%) | | |
| II | 21 | 1 (4.76%) | | |
| Lipodystrophy | 21 | Yes (52.38%) | | |

Data expressed as mean, standard deviation, and confidence interval. BMI, Body mass index; FFA, Fat free mass; cART, Combined antiretroviral therapy; II, Integrase inhibitor; NRTI, Nucleotide reverse transcriptase inhibitor; NNRTI, Non-nucleotide reverse transcriptase inhibitor; PI, Protease inhibitor.

and diastolic blood pressure (DBP) (Table 2). However, in the taVNS condition, MLHIV reported higher pain after the intervention compared to the sham condition (*Md* = 3.04, *IQR* = .00–5.00 vs. *Md* = 0.47; *IQR* = .00 −.00; *p* < .001; *d* = 1.32).

The GEE revealed no significant effects of condition (rMSSD: $\chi^2(1)$ =.049, p = .484; pNN50: $\chi^2(1)$ = 1.160, p = .281), time (rMSSD: $\chi^2(5)$ = 5.81, p = 0.325; pNN50: $\chi^2(5)$ = 6.250, p = .283), or condition-by-time interaction (rMSSD: $\chi^2(5)$ =.260, p = .998; pNN50: $\chi^2(5)$ =.130, p = 1.000) for rMSSD or pNN50, indicating taVNS did not significantly alter these measures compared to sham over time. Effect sizes (Cohen's d) for within-condition changes relative to baseline ranged from moderate to large for rMSSD and pNN50 in taVNS, and from small to moderate (rMSSD) or moderate to large (pNN50) in sham, with similar trends over time. Between-condition effect sizes were consistently small at each time point, with overall effect sizes of d = 0.52 (rMSSD) and d = 0.78 (pNN50), indicating moderate effects. Marginal effects of taVNS versus sham were nonsignificant for rMSSD (2.08 ms, 95% CI: −3.77 to 7.99 ms, Z = 0.69, p = .485) and pNN50 (2.23%, 95% CI: −1.84 to 6.37%, Z = 1.08, p = .282). Descriptive results are detailed in Table 3.

## Discussion

PLHIV may present reduced cardiac vagal activity as a characteristic of the virus' action on the central and autonomic nervous systems. Cardiac vagal activity can be modulated by taVNS which has been used as a non-pharmacological strategy in the treatment of various clinical conditions. This proof-of-concept study is the first controlled investigation to assess the acute effects of taVNS on cardiac vagal activity in MLHIV. Our hypothesis was that acute taVNS could improve HRV indices associated with cardiac vagal activity in men with HIV. However, our results did not confirm this hypothesis. The main findings were: a) rMSSD and pNN50 did not differ between conditions (taVNS vs. sham); b) Over time, rMSSD and pNN50 did not differ in both conditions; and c) rMSSD and pNN50 showed no interaction between condition and time, suggesting that taVNS did not modulate cardiac vagal activity in PLHIV.

rMSSD and pNN50 are commonly used in taVNS studies on HRV, as both variables are suggested to index cardiac vagal activity [51]. In the present study, although rMSSD and pNN50 exhibited a tendency to increase over time (Fig 4), with moderate effect sizes between stimulation conditions (rMSSD: *d* = 0.52; pNN50: *d* = 0.78), these changes were not sufficient to reach statistical significance. This contrasts with the findings of Borges et al. [62,63]. In their study, the authors conducted three experiments investigating different taVNS intensities on a vagally mediated HRV marker and observed an increase in cardiac vagal activity regardless of the stimulation intensity used or the tested condition (taVNS or sham) [62].

Table 2. Intervention characteristics, autonomic, and cardiovascular parameters.

| | taVNS | sham | *p** | Cohens *d* |
|---|---|---|---|---|
| **Intervention parameters** | | | | |
| Perception threshold (mA) | 4.81 (4.00–5.00) | 4.90 (3.00–5.00) | .890* | 0.042 |
| Intensity (mA) | 8.71 (6.00–10.0) | | | |
| Pain | 3.04 (.00–5.00) | 0.47 (.00–.00) | **<.001** | 1.327 |
| **Autonomic and cardiovascular parameters** | | | | |
| rMMSD (ms) | 29.57 (19.4–39.7) | 29.89 (19.6–40.1) | .964$ | 0.013 |
| pNN50 (%) | 10.19 (3.29–17.1) | 8.96 (2.48–15.4) | .792$ | 0.082 |
| SBP (mmHg) | 118.85 (111.00–130.00) | 117.71 (108.00–128.00) | .795* | 0.080 |
| DBP (mmHg) | 73.85 (67.00–84.00) | 76.90 (69.00–87.00) | .422* | 0.250 |

*Student's t-test.

$Generalized Estimated Equation with gamma distribution and data expressed as mean and (95% Coefficient Interval). Other data expressed as mean and interquartile ranges. ms, milliseconds; rMSSD, square root of the mean square of the differences between RR intervals; pNN50 (%), percentage of differences between adjacent normal intervals greater than 50 ms; SBP, systolic blood pressure; DBP, diastolic blood pressure; mmHG, millimeters of mercury; significant p-value in bold.

**Table 3. Comparison of the effects of condition (taVNS and sham), and effects of time on rMSSD and pNN50 in men living with HIV (n = 21).**

| | Baseline | Stimulation | | Recovery | | | P-values of main effects | | Cohen's d |
|---|---|---|---|---|---|---|---|---|---|
| | Moment 1 | Moment 2 | Moment 3 | Moment 4 | Moment 5 | Moment 6 | Time | Condition | Condition |
| **taVNS rMSSD (ms)** | 29.57 | 32.46 | 31.96 | 38.22 | 37.57 | 40.77 | .325 | .484 | 0.52 |
| | (20.82–38.32) | (22.86–42.07) | (22.50–41.42) | (26.91–49.53) | (26.45–48.69) | (28.71–52.84) | | | |
| **Cohen's $d_{RM, pooled}$ Within** | — | 0.67 | 0.48 | 1.46 | 0.84 | 1.53 | | | |
| **Sham rMSSD (ms)** | 29.89 | 29.10 | 31.20 | 34.82 | 34.59 | 38.49 | | | |
| | (21.04–38.74) | (20.49–37.72) | (21.97–40.43) | (24.52–45.13) | (24.35–44.83) | (27.10–49.87) | | | |
| **Cohen's $d_{RM, pooled}$ Within** | — | 0.10 | 0.16 | 0.54 | 0.53 | 0.70 | | | |
| **Mean difference** | −0.32 | 3.36 | 0.76 | 3.40 | 2.98 | 2.28 | | | |
| **Cohen's d Between** | 0.01 | 0.14 | 0.03 | 0.14 | 0.13 | 0.09 | | | |
| **taVNS pNN50 (%)** | 10.19 | 13.44 | 12.48 | 15.19 | 16.40 | 18.67 | .283 | .281 | 0.78 |
| | (4.10–16.29) | (6.37–20.52) | (5.95–19.27) | (7.59–22.80) | (8.43–24.36) | (10.2–27.32) | | | |
| **Cohen's $d_{RM, pooled}$ Within** | — | 0.73 | 0.63 | 0.92 | 0.91 | 1.29 | | | |
| **Sham pNN50 (%)** | 8.96 | 10.10 | 10.88 | 12.85 | 14.04 | 16.16 | | | |
| | (3.24–14.68) | (4.04–16.17) | (4.58–17.17) | (5.95–19.74) | (6.79–21.3) | (8.27–24.05) | | | |
| **Cohen's $d_{RM, pooled}$ Within** | — | 0.47 | 0.72 | 1.00 | 1.01 | 0.99 | | | |
| **Mean difference** | 1.23 | 3.34 | 1.60 | 2.34 | 2.36 | 2.51 | | | |
| **Cohen's d Between** | 0.08 | 0.19 | 0.11 | 0.15 | 0.13 | 0.14 | | | |

Data analyzed using the Generalized Estimated Equation with fixed effects and Gamma Distribution and identify link function. Data expressed as mean and (95% Coefficient interval). Cohen's $d_{RM, pooled}$, Within, dRepeated Measures, pooled was calculated with rest as reference. Cohen's d Between, was calculated between the conditions for each moment. Mean difference, mean of sham minus mean of taVNS. rMSSD, square root of the mean square of the differences between RR intervals; ms, milliseconds; pNN50 (%), percentage of differences between adjacent normal intervals greater than 50 ms. p-value significance < 0.05.

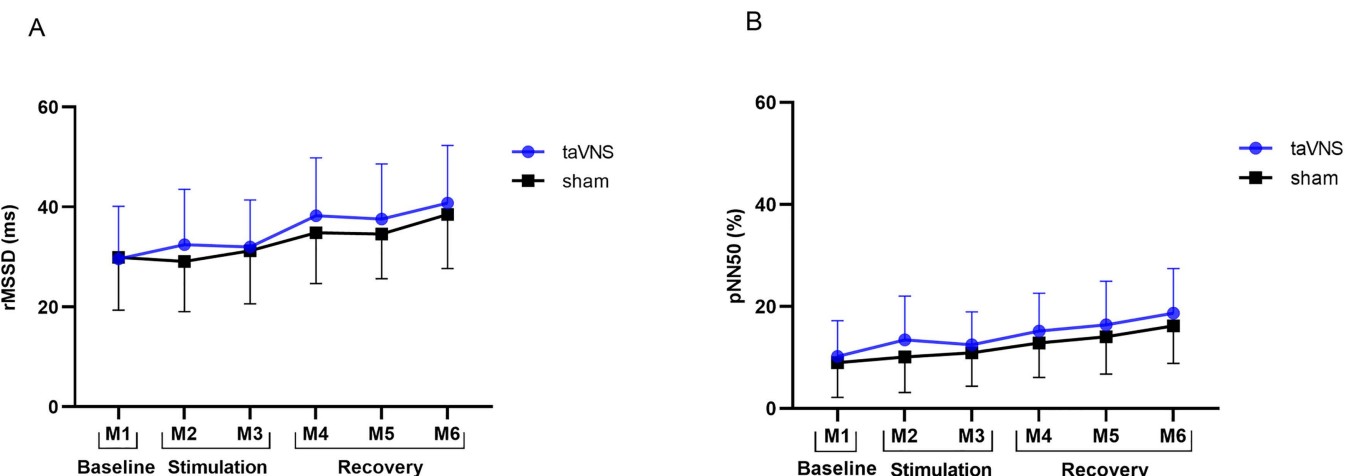

**Fig 4. Behavior over time of the rMSSD (A) and pNN50 (B) variables of men living with HIV (MLHIV) under two stimulation conditions (taVNS and sham).** M, moments; rMSSD, square root of the mean square of the differences between RR intervals; pNN50 (%), percentage of differences between adjacent normal intervals greater than 50 ms.

Conversely, Forte et al. [24] demonstrated that 10 minutes of stimulation were sufficient for taVNS to modulate HRV in young healthy individuals, significantly increasing rMSSD, SDNN, and HF values. The authors attributed these effects to specific protocol parameters, such as the use of an active sham stimulation applied to the auricular helix, which may have reduced potential confounding biases among participants.

Despite the increasing number of studies on taVNS, the sham in controlled studies in this area lacks investigation. The use of the earlobe as an active sham stimulation is common, and the choice of this structure is based on imaging studies [41,64,65], which in turn used the tragus for taVNS, and a dissection study [66]. Peuker and Filler [66] described in detail the nerve distribution of different areas of the auricle from seven human cadavers and presented the earlobe as an area free of vagal innervation. However, it was not possible to prove that electrical stimulation in the earlobe cannot stimulate the central nuclei of the brain that increase cardiac vagal flow [67]. Unlike other studies that used the earlobe as an active sham condition, we opted for a passive sham design, maintaining electrode placement identical to the taVNS condition. This decision aimed to minimize confounding effects, as active sham stimulation in another auricular region could inadvertently stimulate muscular areas with potential vagal influence [68,69]. Nonetheless, a study on pain has already shown that sham passive does not differ from sham active but could generate mechanical pressure exerted by the electrodes, stimulating vagal nerve endings [70]. However, this assertion was only a supposition and has not been empirically supported.

The absence of differences may be influenced by various parameters of the stimulation protocol, including intensity. For intensity adjustment, we followed the recommendations proposed by Badran et al. [43]. Volunteers who reported discomfort had their intensity immediately reduced, but not necessarily below the perceptual threshold, as vagus nerve activation is suggested to occur between the perceptual and pain thresholds [71]. The mean intensity used in this study was 8 mA, which falls within the range of previous taVNS studies [72,73]. For example, Geng et al. [74] reported a mean intensity of 16 mA and found significant differences in rMSSD between conditions (taVNS vs. sham) and in the condition × time interaction; however, taVNS was applied to the tragus. Some studies suggest that higher intensities may enhance neuromodulatory effects [75]. However, it has been shown that higher intensity can lead to greater discomfort, which is not associated with higher values of rMSSD [62], whereas others report significant changes in HRV even at lower intensities (e.g., 1.2 mA) [24]. It is important to note that direct comparisons of stimulation intensities across studies should be interpreted with caution due to methodological variations. Studies report intensities based on device readings, which may not be standardized or directly comparable. Despite some attempts to investigate the use of different stimulation parameters [62], different current waveforms (such as biphasic vs. monophasic) may elicit distinct stimulation sensations for participants, particularly when the stimulation site differs (tragus vs. cymba conchae) [76,77], requiring different current levels for each waveform to achieve a comparable sensory experience. Given these discrepancies, we recommend that future studies conduct more in-depth investigations to systematically explore the interaction between stimulation intensity, waveform characteristics, electrode placement in different ear regions, and, consequently, skin thickness assessment.

It is possible that HRV may not be an ideal biological marker for measuring the modulation of cardiac vagal activity in taVNS studies [78]. The existence of many protocols, and which stimulation parameters are considered ideal, continue to be a subject of discussion in both healthy individuals and clinical groups [23,25,78,79]. Systematic reviews show a variety of results due to different study designs and stimulation parameters. So far, taVNS does not have a significant acute effect on HRV in various scenarios. Although several studies report changes in HRV during taVNS, these alterations are observed only when comparing baseline HRV to HRV during stimulation. However, when comparing HRV between taVNS and sham stimulation in healthy individuals and patients with different clinical conditions, the effect is mostly non-existent, except for the LF/HF index [10,78]. Nevertheless, it is important to highlight that the LF/HF ratio is not an ideal marker of cardiac vagal activity, which limits its interpretability [50,51]. Still, certain patient groups, particularly those with differences in baseline cardiac vagal activity, may exhibit distinct responses to taVNS [78,79].

Being the first study of taVNS in PLWH, comparability is difficult due to the characteristics of the disease. Although we conducted a randomized, crossover clinical trial to reduce sample heterogeneity, it is still possible that in HIV there

are relevant factors that need to be investigated, such as the amount of virus in the nervous system, neural metabolic disorders, which could reduce the responsiveness to taVNS. Furthermore, conducting clinical trials in the HIV domain is challenging [80]. The challenges range from the conception of the trial and recruitment of participants to ethical and operational considerations [81]. These challenges are compounded by the evolving nature of HIV treatment, which requires adaptive trial designs and a deep understanding of the disease's impact on patients [82].

The absence of taVNS effects on cardiac vagal activity may be attributed to the complex mechanisms underlying taVNS activation in the cardiovascular system. taVNS appears to excite afferent fibers that do not directly innervate the heart but may indirectly modulate it via brainstem nuclei [76]. This process enhances the input to the nucleus tractus solitarius (NTS) and influences the activity of NTS neurons that project to vagal cardioinhibitory efferent neurons located in the dorsal vagal nucleus (DVN) and the nucleus ambiguus (NA). These vagal efferent neurons propagate vagal tone to the sinoatrial (SA) node [83]. Additionally, taVNS may also excite NTS neurons that send excitatory projections to the caudal ventrolateral medulla (CVLM), which inhibits the rostral ventrolateral medulla (RVLM) – the primary source of excitatory drive to sympathetic pre-ganglionic neurons in the intermediolateral cell column (IML) of the spinal cord [21]. In addition to this complex communication, people with HIV are predisposed to a variety of different fluid, electrolyte, and acid-base disorders, often silent [84], besides having drier skin than healthy subjects [85]. Such observations might reduce the conduction of the stimulus in the ear [86,87], thus negatively impacting the potential of taVNS for people with HIV.

## Limitations

The main limitation of our study was not assessing pro-inflammatory cytokines such as Interleukin 6 (IL6), Interleukin 1 beta (IL1β), Tumor Necrosis Factor alpha (TNFα), and C-Reactive Protein (CRP), as well as Brain-Derived Neurotrophic Factor (BDNF). People living with HIV may exhibit higher inflammatory markers due to the combined action of the virus and antiretroviral therapy. These effects, isolated or in combination, might lead to a dysregulation in the inflammatory reflex controlled by the vagus nerve [88]. In animal studies, it has been shown that 1) BDNF can inhibit the HIV gp120 protein, helping to reduce microglial and astrocyte infection in the CNS through independent and anti-inflammatory properties [89] and 2) that BDNF expression increased when the vagus nerve was stimulated; however, the experiments were conducted in animals and the stimulation was not transcutaneous [90]. Therefore, we cannot affirm that the investigated volunteers presented a high inflammatory profile and that their BDNF levels were low, as we did not assess this. For this reason, we propose for future taVNS studies using a population with HIV to consider pro-inflammatory cytokines.

Regarding cardiac vagal activity, only two markers were evaluated, rMSSD and pNN50. Other associated physiological markers could have been compared to cardiac vagal activity, such as evoked somatosensory potentials and noradrenergic markers like pupil dilation, P300, and salivary alpha-amylase [77].

Participants in the study spent the entire session seated on a couch, watching POV-style videos of streets in different countries. Care was taken to select videos of the same style and content to avoid evoking emotions such as anger, sadness, or euphoria. However, this aspect was not controlled in the study, as the videos had not been empirically tested or validated as neutral. Finally, regarding pain assessment, at the end of the stimulation, an 11-point numerical scale was presented by an evaluator, who sometimes needed to clarify doubts for the assessed individual on how to interpret it. This fact could be associated with cognitive performance, but no association was made.

## Conclusion

In conclusion, this proof-of-concept study indicates that the acute application of taVNS did not significantly impact cardiac vagal activity in men with HIV. The stimulation parameters used did not produce meaningful differences compared to sham stimulation. While previous studies have reported positive effects of taVNS on rMSSD and pNN50 in healthy individuals, our findings suggest that these effects may not be directly translatable to MLHIV, underscoring the importance of considering population-specific characteristics when assessing taVNS efficacy. Future research should explore

alternative stimulation protocols, including bilateral stimulation, different ear regions, and individual skin properties (e.g., impedance, water content, structure, and subcutaneous fat thickness). Additionally, incorporating complementary physiological markers could provide a more comprehensive understanding of taVNS effects, particularly in clinical populations such as PLHIV. These investigations will be crucial for refining taVNS applications and optimizing its therapeutic potential in diverse health conditions.

## Supporting information

**S1 File. CONSORT checklist.**
(PDF)

## Author contributions

**Conceptualization:** Jason Medeiros, Phelipe Wilde, Rafaela Catherine da Silva Cunha de Medeiros.

**Data curation:** Jason Medeiros, Phelipe Wilde, Amon Gonçalves de Melo Neto.

**Formal analysis:** Phelipe Wilde.

**Funding acquisition:** Paulo Moreira Silva Dantas.

**Investigation:** Jason Medeiros.

**Methodology:** Jason Medeiros, Phelipe Wilde, Rafaela Catherine da Silva Cunha de Medeiros, Júlio César Medeiros Alves, Amon Gonçalves de Melo Neto.

**Project administration:** Jason Medeiros, Júlio César Medeiros Alves, Amon Gonçalves de Melo Neto.

**Resources:** Paulo Moreira Silva Dantas.

**Supervision:** Jason Medeiros, Júlio César Medeiros Alves, Amon Gonçalves de Melo Neto, Jason R Jaggers, Paulo Moreira Silva Dantas.

**Validation:** Uirassu Borges, Phelipe Wilde, Rafaela Catherine da Silva Cunha de Medeiros, Amon Gonçalves de Melo Neto, Paulo Moreira Silva Dantas.

**Visualization:** Júlio César Medeiros Alves.

**Writing – original draft:** Jason Medeiros, Uirassu Borges, Rafaela Catherine da Silva Cunha de Medeiros.

**Writing – review & editing:** Jason Medeiros, Uirassu Borges, Phelipe Wilde, Rafaela Catherine da Silva Cunha de Medeiros, Júlio César Medeiros Alves, Jason R Jaggers, Daniel Gomes da Silva Machado, Ronaldo Vagner Thomatieli dos Santos, Paulo Moreira Silva Dantas.

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
