## [Decision Letter · Decision Letter 0]

11 Sep 2024

Dear Dr. Medeiros,

Thank you for submitting your manuscript to PLOS ONE. After careful consideration, we feel that it has merit but does not fully meet PLOS ONE’s publication criteria as it currently stands. Therefore, we invite you to submit a revised version of the manuscript that addresses the points raised during the review process.

We look forward to receiving your revised manuscript.

Kind regards,

Nejka Potocnik

Academic Editor

PLOS ONE

“This study was partially funded by the Coordination for the Improvement of Higher Education Personnel (CAPES) - Brazil (Funding Code 001), which awarded doctoral scholarships to JAM and PW. https://www.gov.br/capes/pt-br”

Reviewers' comments:

Reviewer's Responses to Questions

**Comments to the Author**

1. Is the manuscript technically sound, and do the data support the conclusions?

Reviewer #1: Yes

Reviewer #2: Partly

Reviewer #3: No

2. Has the statistical analysis been performed appropriately and rigorously?

Reviewer #1: Yes

Reviewer #2: Yes

Reviewer #3: No

3. Have the authors made all data underlying the findings in their manuscript fully available?

Reviewer #1: Yes

Reviewer #2: No

Reviewer #3: No

4. Is the manuscript presented in an intelligible fashion and written in standard English?

Reviewer #1: Yes

Reviewer #2: No

Reviewer #3: Yes

Reviewer #1: Review of PLOS ONE PONE-D-24-18721 manuscript July 2024

‘Acute effect of transcutaneous auricular vagus nerve stimulation on cardiac vagal

activity in men living with HIV: a randomized clinical trial’.

The main aim of this research article is clear. The aim is to test taVNS in HIV persons to see if it affects vagal activity. Apart from improving vagal activity, could taVNS lower sympathetic activity in HIV persons or healthy persons?

The aim is novel, as this data has not been previously collected. The Introduction and Methods section have appropriate detail. Some more details could be added to the Methods.

Results are relatively clear and the data is reasonably well presented with tables and graphs. Although some sections could be better written, especially on GMM data.

The Discussion has a satisfactory level of critical analysis, although more can be added on the time changes, could they be related to a repeated effect. The significance of the study was given, and comparisons made with previous studies, and an emphasis on the applied nature of this research.

They are no major issues with the methodology or data analysis. This manuscript could be better presented. The language and grammar is acceptable, some improvements in grammar could be made. References are correct but a more concise number could be used. Ref 44 and 45 are the same.

Corrections/suggestions for the authors are mentioned below:

Abstract

1. Line 33-34. Clearly state if there was no significant difference in parameters between active and placebo.

Intro

1. Line 58. Should ‘autonomic peripheral system’ be the ‘autonomic nervous system’?

2. Line 60. Add ‘could counteract some of these impairments’

3. Line 83. What type of vagal activity was used?

4. Line 89. Which part of the auricle? The whole external ear? This detail has been given in the methods, but could be added here for greater clarity.

Methods

1. Line 110. MLHIV meaning? Were females not available?

2. Line 153. Why was the left ear used?

3. Line 161. How many subjects had a lowered intensity?

4. Line 164. So was the intensity at 0 mA for placebo?

5. Line 188. Why was the last 10 minutes selected for HRV analysis, and not the first 10 minutes? This seems to contract Lines 196-197?

6. Line 198. State LH, HF and ratio assessed.

7. Line 201. When was BP measured?

8. Line 218. Use consistent terminology experimental/control or active/placebo.

9. Line 221. Could watching a video affect their HRV?

10. Line 225. Line 188 says it was the last 10 minutes?

11. Line 229 and 231. Two Fig 3 captions are given and for the first one remove ‘This is the Fig 3’.

12. Line 257. Virtual or visual for pain scale?

Results

1. Line 264. Height and weight are given in Table 1, but it is not mentioned how this was measured in the Materials section. Same for viral load and cART data.

2. Line 279 Use SDRR in place of SDNN. Are these values not very low? Normally they are within range of 20-80 ms.

3. Line 297-307. These changes could be written more concisely.

4. Line 310 typo.

Discussion

1. Line 324. Use improve in place of increase, as SDRR would decrease if better and likewise ratio would go down, but HF would increase.

2. Line 425. Would this limitation be countered by the fact other PWHIV studies have measured HRV.

3. Line 396, 446, and 454. Typos.

Reviewer #2: Thank you for inviting me to review this paper it is an honour and a privilege to do so.

On the topic of the paper itself, the paper focuses on autonomic neuromodulation through transcutaneous auricular vagus nerve stimulation (taVNS) in individuals living with HIV. The field of electrical modulation of the parasympathetic system (PNS) has experienced significant expansion in recent years due to the non-invasive nature of the procedure, which has little to no side effects. The authors are the first to apply this technique to the population of patients with HIV. They employed a fairly standard approach by conducting a crossover study that included both a sham control and single blinding. Their selected marker for assessing PNS activation was heart rate variability (HRV). This is a reasonable choice, as it is one of the very few indicators that have been shown to change in response to taVNS (when compared to sham), although only in the frequency domain using the LF/HF ratio (Hua et al., 2023).

The study which the paper describes is in itself fine. The quality and execution of the study is in line with other studies in this field. However, there are some key issues that need to be addressed. The introduction and discussion sections are inadequately presented. There should be improved coherence in the discussion of topics. Currently, topics are referenced multiple times in various contexts, at times in ways that are very confusing for the reader.

There are issues regarding blinding. Although the study is presented as single-blinded, the blinding issues are substantial enough that the study, in my view, cannot be classified as such (refer to major issues, line 165 in the manuscript). Furthermore, it is unclear which periods were used for HRV calculations as it is very poorly presented - if different periods were used to calculate HRV for each measurement, which was my understating from reading the methods, than the results are not comparable between participants and interventions (sham and taVNS). If this is indeed the case, the HRV calculations and the statistical analysis should be redone using the exact same five-minute periods for all participants. It may be somewhat acceptable (though not ideal) to choose arbitrary five-minute periods during baseline HRV assessment, but not during the stimulation phases and recovery, where changes in the dynamic of HRV are not only likely, but expected!

To summarize: The study was conducted accurately, but the presentation is flawed, as the patients were aware that one stimulation differed from the other, thus compromising the blinding process. Furthermore the calculation of HRV should be done on the exact same 5 min periods across all measurements. This issue must be rectified prior to publication. There exists a discrepancy in the number of HRV indices presented in the text and in the tables, this has impact on the statistical evaluation, since the correction for multiple testing will vary according to the number of hrv indices used. The presentation requires improvement, as there are moments when it is unclear what the authors intended to convey to the audience.

Major issues:

165 - Consideration of the reviewer¬; regarding sham stimulation: Sham stimulation is a tricky one in the realm of taVNS and I do understand the struggle, there is no ideal solution. Despite that, two limitations of the sham design in this study need to be addressed. 1) The first consideration stems from the fact that during the sham intervention the participants received a tiny amount of stimulation from when their personal stimulation current was being determined, which may or may not have an impact on the PNS. 2) The second consideration is related to the fact that participants were told that the intensity would be lowered and the experience of stimulation would be decreased during the sham intervention but they weren’t told the same during the taVNS intervention. This way the participants received information on what to expect before the stimulation. Additionally if they were briefed beforehand on receiving sham and taVNS on different days they could very easily deduce which is which.

195 – Author: “The 5-minute periods with the most stable heart rate were used for analysis” Consideration of the reviewer: I am not entirely sure about the length of periods used for HRV calculations. Typically, HRV calculations are performed over 5-minute periods. However, in line 195, you refer to 5-minute periods, whereas in line 197, you indicate 10-minute periods. My best guess is that within a 10 minute window an arbitrary 5 minute period was used for HRV calculation that best met some standard. If that is the case, it is quite problematic since HRV calculations are then not comparable between participants and interventions.

Table 2 – Consideration of the reviewer: Up until now there was no mention of other than RMSSD and pNN50 HRV indices, but in this table others are listed as well (SDNN, HF, LF, LF/HF)! If indeed these indices were calculated (based on the fact that statistics for them were presented in the aforementioned table) a correction for multiple testing needs to be done in the statistical analysis.

Minor issues:

From the abstract – Consideration of the reviewer: The usual nomenclature for a control group in a study focused on electrical stimulation is sham and not placebo, it can be either active sham (electrical current is passed through the test subject) or passive/inactive sham (no current is passed through the test subject)

From the abstract – Consideration of the reviewer: Better than using “rest” as a way of addressing baseline readings, just use baseline, since the reader immediately knows that you are referring to the period before the stimulation took place, rest could also equally be used for recovery so exactly defining both baseline and recovery is good practice.

59-62 Consideration of the reviewer: The sentence does not tell the reader what the “new non-pharmacological adjunct therapy” is, you are probably referring to taVNS, but you should define it, also a source is needed for the whole statement.

72. Author: “endothelial cells that form the blood vessels” Consideration of the reviewer: This information is redundant since the endothelium only builds the inner wall of vessels.

74. Author: “Furthermore, peripheral nerve fibres can be directly infected by HIV, and through efferent communication, they can reach the central nervous system”. Consideration of reviewer: Did you have retrograde axonal transport in mind? Efferent is the process of going further from the source (in this case the central nervous system) and afferent coming towards the source. Communication in the nervous system is done via electrical (action or graded) potentials, material of any kind cannot be transported in this manner.

78-82 Author: “Moreover, HIV can induce oxidative stress in the central nervous system, resulting in abnormal signals from sensory receptors” . Consideration of reviewer: Surely oxidative stress in the CNS cannot result in peripheral receptors producing abnormal signals?

82-84. Author: “Lower resting HRV indices in people with HIV has been associated with vagal activity compared to people without the HIV”. Consideration of reviewer: Probably meant “lower vagal activity”?

85-87 Author: “Thus, non-pharmacological therapies focusing on autonomic adjustments become necessary and essential for reducing cardiovascular risks as is the case with taVNS”. Consideration of reviewer: Something that hasn’t been proven to be beneficial and you are researching for the first time on a specific population (in your case HIV patients) can’t really be essential, it can be potentially beneficial though.

91 Author: “According to the neurovisceral integration model, the prefrontal cortex regulates cardiac function through cardiac vagal activity” – Source needed.

97. Author: “However, despite major impairment of cardiovascular activity caused by HIV…” - Source needed.

102-106 Consideration of reviewer: The end of the introduction deserves a more concrete segue into highlighting what the scientific problem is and how exactly does the study that the authors conducted address it.

193 Author: For HRV analysis, low-level artifact correction was applied, the sample length was set at 5 minutes for the HRV process, and a 2% threshold was used for beat corrections. Consideration of reviewer: As someone who has not used Kubios before I do not know what low level artefact correction is and what “2% threshold for beat correction” is, you will need to clarify this for the reader.

243. Consideration of reviewer: Since you used a generalized linear mixed effect model, I am guessing the distribution of residuals was not normal? It should be mentioned why you chose to use a generalized and not a general mixed effect model. Why did you pick gamma distribution? What was the random effect you mentioned, probably patient, but this should be stated.

294 Author: “From the second recovery measurement (30th min) to rest…” Consideration of reviewer: Implies you only used a single minute of RR intervals to calculate HRV.

344 Author: “The use of the earlobe as an active simulated taVNS stimulation is common, and the choice of this structure is based on imaging studies [46–48] which in turn used the tragus for taVNS, and a dissection study [49].” In this section you address the control as “active simulated taVNS”, further in the manuscript you address it as “simulated stimulation” (line 355) while in the beginning it was “placebo”. The correct terminology is passive sham or active sham.

346 Author: “Peuker e Filler”. Consideration of reviewer: Did you mean “Peuker and Filler”?

358 Author: “We can suppose that the absence of differences was due to the choice of other parameters involved in the stimulation protocol, such as intensity. Consideration of reviewer: It is not really scientific to assume what the cause of some effect was in a study that isn’t ours. We can state the facts and hypothesize, but never assume.

365 - 378 Consideration of reviewer: This section focuses on the comparison of stimulation intensities across various studies. Stimulation intensities are not directly comparable among studies due to the significant variations in measurement methods; most researchers typically depend on the stimulator's display readout, which isn’t a standardized or accurate measurement. Consequently, the reported average stimulation currents vary among studies. Furthermore, even if all studies employed precise current sensing probes, the differing current waveforms (such as biphasic versus monophasic) would elicit a different sensation of stimulation for participants, necessitating different currents for each waveform to achieve a somewhat similar experience.

387 Authors: “Studies have shown relaxing effects of virtual environments, such as watching nature films, on the autonomic nervous system, measured by heart rate variability[60]." Consideration of reviewer: Just a thought, if displaying visual content from a natural environment produces comparable results to taVNS, maybe an intervention using virtual reality and natural landscapes would make sense for a future study.

388 Authors: It is possible that HRV may not be an ideal biological marker for measuring the modulation of cardiac vagal activity in taVNS studies[61]. Consideration of reviewer: Very valid and important point.

399 Authors: “Although we conducted a randomized, crossover clinical trial to reduce sample heterogeneity, it is still possible that in HIV there are relevant factors that need to be investigated, such as the amount of virus in the nervous system…” Consideration of reviewer: this is a very important consideration I am glad that you mentioned it.

406 Authors: “The absence of significant differences of taVNS on HRV can be explained by the first activation pathway, where taVNS operates in an indirect and complex system of brain regions and nuclei[65]”. Consideration of reviewer: Did you mean the absence of effect of taVNS on HRV? What is the first activation pathway, I tried finding it in the source you provided but was unsuccessful.

451. Authors: “Unlike some previous studies that reported positive effects of taVNS on rMSSD and pNN50 in healthy individuals” …” Consideration of reviewer: This must be understood in context; although numerous studies indicate a change in HRV during taVNS, this is only accurate when comparing baseline HRV to HRV during taVNS. When comparing HRV during taVNS against sham in both healthy individuals and patients with different conditions, the effect is mostly non-existent, except for the LF/HF HRV index, which does show change. (Hua et al., 2023; Wolf et al. 2021)

Table 1. Consideration of reviewer: Nicely presented table.

Figure 3. Consideration of reviewer: This figure needs additional information on the x axis. As I understand it, the time stamp indicators on the X axis represent periods, yet in their current form, they just appear as moments in time. Additionally, the repetition of these markers adds to the confusion. The x axis also needs to be relabeled to ECG or RR since that is the signal that was being measured. HRV is a calculated index not a measurement.

Hua, K., Cummings, M., Bernatik, M., Brinkhaus, B., Usichenko, T., & Dietzel, J. (2023). Cardiovascular effects of auricular stimulation -a systematic review and meta-analysis of randomized controlled clinical trials. Frontiers in Neuroscience, 17(September), 1–18. https://doi.org/10.3389/fnins.2023.1227858

Wolf, V., Kühnel, A., Teckentrup, V., Koenig, J., & Kroemer, N. B. (2021). Does transcutaneous auricular vagus nerve stimulation affect vagally mediated heart rate variability? A living and interactive Bayesian meta-analysis. Psychophysiology, 58(11). https://doi.org/10.1111/psyp.13933

Reviewer #3: General comments:

This study evaluated a randomized trial of transcutaneous auricular vagus nerve stimulation (taVNS) on cardiac vagal activity in people living with HIV.

I found a number of methodological points for which I have concerns and have listed those in the specific comments below. I recommend giving some thought to how the methods are explained and how the results are reported, because I found both confusing and somewhat lacking.

Overall, I feel like 21 participants, even in a crossover trial, is probably not enough to achieve robust results. Thus, I think you should lean more heavily toward labeling this a proof-of-concept study rather than expecting this to demonstrate results that are more definitive than that. I feel as though the authors do this a little bit in the discussion, but I would have preferred a stronger lean in that direction.

Specific comments:

1. (lines 111-117) I am confused about how the power analyses are reported here. For almost any trial, power analyses are a required and necessary step. Extenuating circumstances happen and sometimes power and sometimes the desired sample sizes are not met. Sure, that's suboptimal, but it's ok. I don't understand why an after-the-fact power calculation is reported instead. Post-hoc or retrospective power has some controversy and there are divided opinions on the worth of post-hoc power. I think post-hoc power can be used to emphasize end of study conclusions, but I don't find this use of post-hoc power appropriate. Please include the pre-study power analyses and use the section to discuss why the sample size was not met. I think you partially do this on lines 130-135.

2. (lines 239-241) I strongly encourage the authors to use standardized mean differences instead of p-values to compare baseline and 12-month characteristics. Significance testing in these situations is generally frowned upon because a non-significant p-value does not indicate that groups are the same. For info on the topic in relation to baseline imbalance in randomized trials see Altman, https://doi.org/10.2307/2987510 and Senn, https://doi.org/10.1002/sim.4780131703. My recommendation is to use standardized difference to assess differences (see Austin, https://doi.org/10.1080/03610910902859574).

3. (lines 241-245) Without a methodological citation, I am not sure what statistical method is being used. I presume you have run a generalized linear mixed model (GLMM) with because you have assumed a Gamma distribution. But, you said that you've used an identity link function despite the canonical link function for a Gamma being a log link. Please describe in further detail the methods used here with methodological citations and indicate why you have chosen not to use the canonical link function.

4. (line 243) I don't know what a "non-random covariance matrix" is. Again, please provide a methodological citation for this so I can better understand the method implemented.

5. (line 291) I don't recall information in the methods section regarding a mediation analysis. How did you evaluation mediation?

6. (line 293 and onward) What is the interpretation of these effects? Are they mean differences between the treatment conditions? Are they differences in the slopes? If this is a Gamma regression, then maybe they are arithmetic mean ratios? Apologies, but without units on these effect sizes nor description, I am not able to interpret these numbers.

**Do you want your identity to be public for this peer review?** For information about this choice, including consent withdrawal, please see our Privacy Policy

Reviewer #1: **Yes: ** Mirza M F Subhan

Reviewer #2: No

Reviewer #3: No

---

## [Author Response · Author response to Decision Letter 1]

11 Mar 2025

Rebuttal letter

Dear Editor,

Dr. Emily Chenette

PLOS ONE

We are pleased to present here our revised version of the manuscript now entitled “Acute effect of transcutaneous auricular vagus nerve stimulation on cardiac vagal activity in men living with HIV: a proof-of-concept clini-cal trial.” (PONE-D-24-18721.R1) which was submitted to PONE.

The authors would like to thank the Journal and its Editorial Board for allowing us to proceed with the review process. We would like to thank the reviewers for their positive feedback and detailed comments and suggestions on the manuscript. They have identified important aspects of our paper that required relevant amendments, and their evaluations have greatly improved the quality of the manuscript.

We have carefully reviewed all the reviewers’ suggestions and provided a detailed, point-by-point response to each comment. To facilitate the review process, we have highlighted the reviewers’ comments in bold, our responses in blue, and any additions or modifications to the main manuscript in yellow.

Response structure:

• Reviewer’s comment

• Our response

• Added or modified text in the manuscript

In the main manuscript file, all changes have also been highlighted in yellow for easy identification.

Yours sincerely,

The authors.

REVIEWER 1

Reviewer #1: Review of PLOS ONE PONE-D-24-18721 manuscript July 2024

‘Acute effect of transcutaneous auricular vagus nerve stimulation on cardiac vagal

activity in men living with HIV: a randomized clinical trial’.

The main aim of this research article is clear. The aim is to test taVNS in HIV persons to see if it affects vagal activity. Apart from improving vagal activity, could taVNS lower sympathetic activity in HIV persons or healthy persons?

The aim is novel, as this data has not been previously collected. The Introduction and Methods section have appropriate detail. Some more details could be added to the Methods.

Results are relatively clear and the data is reasonably well presented with tables and graphs. Although some sections could be better written, especially on GMM data.

The Discussion has a satisfactory level of critical analysis, although more can be added on the time changes, could they be related to a repeated effect. The significance of the study was given, and comparisons made with previous studies, and an emphasis on the applied nature of this research.

There are no major issues with the methodology or data analysis. This manuscript could be better presented. The language and grammar is acceptable, some improvements in grammar could be made. References are correct but a more concise number could be used. Ref 44 and 45 are the same.

Response: Dear reviewer, the authors would like to thank you for all the invaluable time and energy you devoted to revising the present manuscript as well as for seeing its potential contribution. We have addressed all your comments and suggestions point-by-point below and have amended the text to accommodate your suggestions. All changes are marked in yellow in the main manuscript. We hope we have met your expectations and that the paper is now deemed suitable for publication. The duplicate references have been removed throughout the text.

Corrections/suggestions for the authors are mentioned below:

Abstract

1. Line 33-34. Clearly state if there was no significant difference in parameters between active and placebo.

Response: Thank you very much for the suggestion. We have clarified the absence of differences. The sentence now reads as follows:

“No significant changes in vmHRV parameters were observed over time or between conditions. These findings suggest that an acute taVNS session does not modulate cardiac vagal activity in people living with HIV.” (Abstract, L 33-35).

Intro

1. Line 58. Should ‘autonomic peripheral system’ be the ‘autonomic nervous system’?

Response: Thank you for spotting this erroneous nomenclature. We change the text to the correct nomenclature.

2. Line 60. Add ‘could counteract some of these impairments’

Response: Thank you very much! Another reviewer also suggested a change in this paragraph. They asked to make it clear which therapy we were referring to. After making the revisions, the text now reads as follows:

“These effects may exacerbate the impairments caused by HIV's direct impact on the central nervous, autonomic, and cardiovascular systems, thereby increasing the risk of developing cardiovascular disease [1,6–9]. Transcutaneous Auricular Vagus Nerve Stimulation (taVNS) is a non-pharmacological adjunct therapy with the potential to help mitigate some of the long-term adverse effects of antiretroviral therapy, particularly regarding cardiac autonomic function) [10].” (Intro, L 56-62).

3. Line 83. What type of vagal activity was used?

Response: Dear reviewer, we used cardiac vagal activity. After your observation, we amended the text as follows:

“In accordance, lower resting HRV indices in people with HIV has been associated with lower cardiac vagal activity compared to people without the HIV [19,20].” (Intro, L 82-84).

4. Line 89. Which part of the auricle? The whole external ear? This detail has been given in the methods, but could be added here for greater clarity.

Response: We used the cymba conchae. We have added this information to the text as follows:

“The mechanism of action of taVNS on cardiac vagal activity can be explained by the presence of vagal nerve endings in the human auricle, especially in the regions of the cymba conchae and tragus.” (Intro, L 88-90).

Methods

1. Line 110. MLHIV meaning? Were females not available?

Response 1: Thank you for your comment. The meaning of MLHIV is: Men living with HIV. We included the meaning of the acronym in the text as follows:

“Given that taVNS may positively influence vagal tone, this study aimed to investigate the acute effect of taVNS on cardiac vagal activity in men living with HIV (MLHIV).” (Intro, L 105-107).

As for the inclusion of males’ participants only, we have also added the reasons to the manuscript as follows:

“The study was restricted to male participants for three reasons: First, sex differences in heart rate variability are evident and this could make the results confusing [33]. Second, there is a difference in the autonomic response between the sexes during electrical stimulation of the vagus nerve [34]. Third, women can show fluctuations in HRV, with notable changes between the follicular and luteal phases, mainly due to variations in progesterone levels [35,36].” (Methods, L 114-119).

REFERENCES

Koenig, J., & Thayer, J. (2016). Sex differences in healthy human heart rate variability: A meta-analysis. Neuroscience & Biobehavioral Reviews, 64, 288-310. https://doi.org/10.1016/j.neubiorev.2016.03.007.

Veiz, E., Kieslich, S., Staab, J., Czesnik, D., Herrmann-Lingen, C., & Meyer, T. (2021). Men Show Reduced Cardiac Baroreceptor Sensitivity during Modestly Painful Electrical Stimulation of the Forearm: Exploratory Results from a Sham-Controlled Crossover Vagus Nerve Stimulation Study. International Journal of Environmental Research and Public Health, 18. https://doi.org/10.3390/ijerph182111193.

Schmalenberger, K., Eisenlohr-Moul, T., Jarczok, M., Eckstein, M., Schneider, E., Brenner, I., Duffy, K., Schweizer, S., Kiesner, J., Thayer, J., & Ditzen, B. (2020). Menstrual Cycle Changes in Vagally-Mediated Heart Rate Variability Are Associated with Progesterone: Evidence from Two Within-Person Studies. Journal of Clinical Medicine, 9. https://doi.org/10.3390/jcm9030617.

Schmalenberger, K., Eisenlohr-Moul, T., Würth, L., Schneider, E., Thayer, J., Ditzen, B., & Jarczok, M. (2019). A Systematic Review and Meta-Analysis of Within-Person Changes in Cardiac Vagal Activity across the Menstrual Cycle: Implications for Female Health and Future Studies. Journal of Clinical Medicine, 8. https://doi.org/10.3390/jcm8111946.

2. Line 153. Why was the left ear used?

Response: Thank you for your comment. We decided to stimulate the left ear for some reasons that were also added to the text after your comment. Please see the new sentence of the manuscript below:

“The choice of the left ear for auricular stimulation of the vagus nerve was supported by anatomical and functional evidence indicating a greater density of innervation, significant neuroprotective effects and greater consistency in the therapeutic response [21,41,42]. In addition, for greater comparability we followed the consensus document by Farmer et al (2021) [11] which highlights the predominance of stimulation protocols in the left ear.” (Methods, L 168-173).

REFERENCES

Butt, M., Albusoda, A., Farmer, A., & Aziz, Q. (2019). The anatomical basis for transcutaneous auricular vagus nerve stimulation. Journal of Anatomy, 236. https://doi.org/10.1111/joa.13122.

Yakunina, N., Kim, S., & Nam, E. (2017). Optimization of Transcutaneous Vagus Nerve Stimulation Using Functional MRI. Neuromodulation: Technology at the Neural Interface, 20. https://doi.org/10.1111/ner.12541.

Keute, M., Machetanz, K., Berelidze, L., Guggenberger, R., & Gharabaghi, A. (2021). Neuro-cardiac coupling predicts transcutaneous auricular vagus nerve stimulation effects. Brain Stimulation, 14, 209-216. https://doi.org/10.1016/j.brs.2021.01.001.

Farmer, A., Strzelczyk, A., Finisguerra, A., Gourine, A., Gharabaghi, A., Hasan, A et al. (2021). International Consensus Based Review and Recommendations for Minimum Reporting Standards in Research on Transcutaneous Vagus Nerve Stimulation (Version 2020). Frontiers in Human Neuroscience, 14. https://doi.org/10.3389/fnhum.2020.568051.

3. Line 161. How many subjects had a lowered intensity?

Response: Of the 21 participants, 4 had a reduction in intensity, 1 participant had a reduction up to the perception threshold. Another 3 participants had intensities above the perception threshold by 118%, 125% and 162%. In the sham condition, 2 participants requested a reduction. The sentence now reads as follows:

“In both conditions (taVNS and sham), the researcher remained beside the stimulator throughout the session to intervene in case any participant reported discomfort. During the taVNS condition, intensity adjustments were made for four participants who reported discomfort: one participant had the intensity reduced to the perceptual threshold, while the other three had the intensity adjusted to levels still above the perceptual threshold (118%, 125%, and 162%, respectively). In the sham condition, although the stimulator was turned off, two participants requested a reduction in intensity because they believed they were feeling stimulation. (Methods, L 200-207).

4. Line 164. So was the intensity at 0 mA for placebo?

Response: Exactly. We decided to adopt this sham protocol for the reasons explained below, which was included into the main manuscript:

“The choice of the sham protocol is in line with previous studies that used the same ear region as the active condition to position the electrodes while the electrostimulator remained switched off [42,44,45]. There are protocols that use the ear lobule as an active sham [11] However, it is possible that stimulation in this region may not be entirely specific and could influence multiple neural pathways, leading to activation patterns similar to taVNS stimulation [21].” (Methods, L 194-199).

REFERENCES

Keute, M., Machetanz, K., Berelidze, L., Guggenberger, R., & Gharabaghi, A. (2021). Neuro-cardiac coupling predicts transcutaneous auricular vagus nerve stimulation effects. Brain Stimulation, 14, 209-216. https://doi.org/10.1016/j.brs.2021.01.001.

Couck, M., Cserjési, R., Caers, R., Zijlstra, W., Widjaja, D., Wolf, N., Luminet, O., Ellrich, J., Ellrich, J., Gidron, Y., & Gidron, Y. (2017). Effects of short and prolonged transcutaneous vagus nerve stimulation on heart rate variability in healthy subjects. Autonomic Neuroscience, 203, 88-96. https://doi.org/10.1016/j.autneu.2016.11.003.

Clancy, J., Mary, D., Witte, K., Greenwood, J., Deuchars, S., & Deuchars, J. (2014). Non-invasive Vagus Nerve Stimulation in Healthy Humans Reduces Sympathetic Nerve Activity. Brain Stimulation, 7, 871-877. https://doi.org/10.1016/j.brs.2014.07.031.

Farmer, A., Strzelczyk, A., Finisguerra, A., Gourine, A., Gharabaghi, A., Hasan, A et al. (2021). International Consensus Based Review and Recommendations for Minimum Reporting Standards in Research on Transcutaneous Vagus Nerve Stimulation (Version 2020). Frontiers in Human Neuroscience, 14. https://doi.org/10.3389/fnhum.2020.568051.

Butt, M., Albusoda, A., Farmer, A., & Aziz, Q. (2019). The anatomical basis for transcutaneous auricular vagus nerve stimulation. Journal of Anatomy, 236. https://doi.org/10.1111/joa.13122.

5. Line 188. Why was the last 10 minutes selected for HRV analysis, and not the first 10 minutes? This seems to contract Lines 196-197?

Response: Thank you for the observation. We revised the text to improve its clarity. Additionally, we decided to modify the figure illustrating the HRV analysis moments to clearly indicate the exact time points where RR intervals were recorded and the segments selected for heart rate variability analysis. The updated text and picture can be seen below:

“A total of six 10-minute segments were extracted from the timeline for analysis: Moment 1 (baseline), Moments 2 and 3 during stimulation (0-10 min and 20-30 min), and Moments 4, 5, and 6 during recovery (30-40 min, 50-60 min, and 80-90 min). Within each 10-minute window corresponding to these moments, HRV was analyzed in the first 5 minutes across both conditions and for all participants to ensure better data comparability.” (Methods, L 249-253).

Fig 3 Timeline of recording and analysis of RR intervals. HRVA, heart rate variability analysis; PT, perception threshold; ER, earphone removal; VAS, visual analog scale.

6. Line 198. State LH, HF and ratio assessed.

Response: Dear reviewer, after considering the feedback from another reviewer, we decided to remove the analyses of other time-domain and frequency-domain measures from the tables, as they are not part of the primary outcome. We clarify that our intent was solely to investigate the variables rMSSD and pNN50, as they better represent cardiac vagal activity. The statement is supported by the references below.

REFERENCES

Laborde, S., Mosley, E., & Thayer, J. (2017). Heart Rate Variability and Cardiac Vagal Tone in Psychophysiological Research – Recommendations for Experiment Planning, Data Analysis, and Data Reporting. Frontiers in Psychology, 8. https://doi.org/10.3389/fpsyg.2017.00213.

Malik, M., Camm, A.J., Bigger, J.T., Breithart, G., Cerutti, S., Cohen, R., 1996. Heart rate variability: standards of measurement, physiological interpretation and clinical use. Task force of the European Society of Cardiology and the north American Society of Pacing and Electrophysiology. Circulation 93 (5), 1043–1065.

7. Line 201. When was BP measured?

Response: Thank you for the observation. Blood pressure was measured at baseline. We have clarified this in the text. The revised sentence can be read below:

“Systolic and diastolic blood pressures (SBP and DBP) were measured after 10 minutes of rest at baseline using an Ambulatory Blood Pressure Monitor CONTEC® ABPM50 (Contec Medical Systems - Hebei, China).” (Methods, L 257-259).

8. Line 218. Use consistent terminology experimental/control or active/placebo.

Response: Thank you very much. Following your observation, we have adopted the nomenclature taVNS (active) and sham throughout the text.

9. Line 221. Could watching a video affect their HRV?

Response: Thank you for your observation. We used “point of view” videos to distract participants during the stimulation and sham periods. Please note that being seated looking at a wall for long periods might be boring/tedious and could also direct their attention towards experiment details. We took care to present videos of the same style that would not evoke emotions capable of influencing HRV. Therefore, we did not expect any changes. However, after considering your observation, we recognized this as a limitation (please, see below), as we did not use empirically tested/validated videos as

---

## [Decision Letter · Decision Letter 1]

5 Jun 2025

Acute effect of transcutaneous auricular vagus nerve stimulation on cardiac vagal activity in men living with HIV: a proof-of-concept clinical trial.

PONE-D-24-18721R1

Dear Dr. Medeiros,

We’re pleased to inform you that your manuscript has been judged scientifically suitable for publication and will be formally accepted for publication once it meets all outstanding technical requirements.

Kind regards,

Mehmet Demirci, PhD

Academic Editor

PLOS ONE

Additional Editor Comments (optional):

Reviewers' comments:

Reviewer's Responses to Questions

**Comments to the Author**

Reviewer #1: All comments have been addressed

Reviewer #4: All comments have been addressed

Reviewer #5: All comments have been addressed

2. Is the manuscript technically sound, and do the data support the conclusions?

Reviewer #1: Yes

Reviewer #4: Yes

Reviewer #5: Yes

3. Has the statistical analysis been performed appropriately and rigorously?

Reviewer #1: Yes

Reviewer #4: Yes

Reviewer #5: Yes

4. Have the authors made all data underlying the findings in their manuscript fully available?

Reviewer #1: Yes

Reviewer #4: Yes

Reviewer #5: No

5. Is the manuscript presented in an intelligible fashion and written in standard English?

Reviewer #1: Yes

Reviewer #4: Yes

Reviewer #5: Yes

Reviewer #1: Replies to reviewer comments are appropriate. Thank you for the detailed replies to all my queries and making corrections as mentioned.

Reviewer #4: The acute effects of taVNS on cardiac vagal activity in men with HIV are a novel and clinically relevant question that is addressed in this manuscript. The authors deserve praise for their thorough and open approach, especially in their handling of reviewer comments. The study's rigor has increased due to noteworthy methodological clarifications, such as better blinding control as well as uniform HRV time segment selection.

Even though the results were null, they are still significant for early-stage studies. The discussion is well-considered and places the findings in the larger body of literature. Future research directions are made clear, and limitations are openly acknowledged.

I am in favor of this updated manuscript being published.

Reviewer #5: The authors have addressed the point raised by the reviewers. The concept of using PLWHIV with this is not clear to me in terms of other viruses might also induced such changes and the significance of this with disease outcomes? It seems like an restorative therapy for curbing the long term effect of antivirals and so should be framed as such. This is what is being evaluated , the current title makes it seem it has something to do with HIV treatment per se.

**Do you want your identity to be public for this peer review?** For information about this choice, including consent withdrawal, please see our Privacy Policy

Reviewer #1: **Yes: ** Mirza M F Subhan

Reviewer #4: No

Reviewer #5: No

---

## [Editor Report · Acceptance letter]

PONE-D-24-18721R1

PLOS ONE

Dear Dr. Medeiros,

I'm pleased to inform you that your manuscript has been deemed suitable for publication in PLOS ONE. Congratulations! Your manuscript is now being handed over to our production team.

Kind regards,

on behalf of

Assoc. Prof. Mehmet Demirci

Academic Editor

PLOS ONE